# Improving Biodiversity Offset Schemes through the Identification of Ecosystem Services at a Landscape Level

Annaêl Barnes [1,2,3,*], Alexandre Ickowicz [1,2], Jean-Daniel Cesaro [1,2], Paulo Salgado [1,2], Véronique Rayot [4], Sholpan Koldasbekova [5] and Simon Taugourdeau [1,2]

1 CIRAD, UMR SELMET, F-34090 Montpellier, France
2 UMR SELMET, University of Montpellier, CIRAD, INRAE, Institut Agro, F-34090 Montpellier, France
3 UMR AMAP, University of Montpellier, CIRAD, CNRS, INRAE, IRD, F-34090 Montpellier, France
4 Orano Mining, 125 Av. de Paris, 92320 Châtillon, France
5 KATCO JV LLP, Sauran Street 48, Astana 020000, Kazakhstan
* Correspondence: annael.barnes@cirad.fr

**Abstract:** Biodiversity offsets aim to compensate the negative residual impacts of development projects on biodiversity, including ecosystem functions, uses by people and cultural values. Conceptually, ecosystem services (ES) should be considered, but in practice this integration rarely occurs. Their consideration would improve the societal impact of biodiversity offsets. However, the prioritisation of ES in a given area is still limited. We developed a framework for this purpose, applied in rangelands landscapes in Kazakhstan, in the context of uranium mining. We assumed that different landscapes provide different ES, and that stakeholders perceive ES according to their category (e.g., elders and herders) and gender. We performed qualitative, semi-structured interviews with a range of stakeholders. Using the Common International Classification of Ecosystem Services, we identified 300 ES in 31 classes across 8 landscape units. We produced a systemic representation of the provision of ES across the landscapes. We showed a significant link between ES and landscape units, but not between ES and stakeholder categories or gender. Stakeholders mostly identified ES according to the location of their villages. Therefore, we suggest that the biodiversity offsets should target ES provided by the landscape unit where mining activities occur and would be most interesting in the landscapes common to all villages. By performing a systemic representation, potential impacts of some offset strategies can be predicted. The framework was therefore effective in determining a bundle of ES at a landscape scale, and in prioritising them for future biodiversity offset plans.

**Keywords:** ecosystem services; biodiversity offset; CICES; landscape units; stakeholders; rangelands

## 1. Introduction

Biodiversity loss and ecosystem degradation are massive environmental problems worldwide [1,2]. To mitigate biodiversity losses due to development projects (such as mining activities), policy makers, governments and the private sector are increasingly adhering to biodiversity offset mechanisms [2–6]. Biodiversity offsets are defined as 'measurable conservation outcomes resulting from actions designed to compensate significant residual adverse biodiversity impacts arising from project development after appropriate prevention and mitigation measures has been taken' [7]. These mechanisms target the residual impacts that cannot be avoided, reduced, and will not be restored, according to the steps of the mitigation hierarchy [2,5,6]. Biodiversity offsets have a goal of no net loss (NNL) of biodiversity or when possible, a net gain (NG) of biodiversity [2,7,8], including species composition, habitat structure, ecosystem function, its use by people and associated cultural value [7]. The methods for implementing offsets are diverse: restoration of degraded ecosystems, creation of new habitats, protection of existing high quality ecosystems at risk of degradation or loss, change of practice in favour of biodiversity on already managed areas, or mitigation bank [3,7,9–11].

Ecosystem services (ES) are defined as 'the aspects of ecosystems utilized (actively or passively) to produce human well-being' [12]. Benefits derive from ES and are 'the contributions to aspects of well-being' (e.g., health) [13]. The conceptualization of the benefits provided by nature has been driven by a loss of ES [14], mostly resulting from fragmentation and loss of habitat after economic development projects [11]. Several classification systems for describing ES exist, for example, the Millennium Ecosystem Assessment [1], the Intergovernmental Science-Policy Platform on Biodiversity and Ecosystem Services (IPBES, [15]) and also the Common International Classification of Ecosystem Services (CICES) [16,17], which has become a common international reference [18,19]. Its hierarchical structure (from least to most detailed: section, division, group and class of ES) allows users to go to the level of detail wanted or needed [16,17]. The CICES can also include abiotic flows, which may be socially important [19], while abiotic structure and processes are less or not explicitly integrated in some other classification frameworks [19,20].

Although it can be argued that biodiversity provides ES and needs them for its persistence [7,21], the relationship between biodiversity and ES provision is not so obvious. Some ES can be provided by a diverse set of species and habitats, while others are strongly linked to specific species or sets of species (e.g., pollination) [22]. Moreover, area considered as important for biodiversity conservation will not necessarily be crucial for ES supply and vice versa [11]. In a study in Iran, Karimi et al. [23] showed a strong relationship between cultural services and biodiversity hotspots, but found a weaker link between provisioning services and biodiversity. Therefore, the relationship between biodiversity and ES provision depends on the ES considered.

From a conceptual perspective, biodiversity offset schemes could consider how changes in biodiversity might influence the provision of ES to different types of stakeholder [7,10,21]. Some methods for integrating ES into biodiversity offsets have already been developed [7], but current offsetting is mostly focused on critical habitats and threatened species [5], rather than on ordinary biodiversity and the services it provides [5,22]. The failure to take proper account of ES is due to several reasons. There is no standardized and systematic methodology for the integration of ES into the mitigation hierarchy [22], or biodiversity offset schemes [5]. Coupled with a lack of legislation for the consideration of ES in offsets programs [24], and Environmental and Social Impact Assessment (ESIA) of economic development projects [11], ES are rarely mentioned in existing offset practices [5], or are a consequence of the chosen offset strategy and not a driver [24]. Nevertheless, integrating ES into offset mechanisms is receiving increasing attention in the international community in recent years [5,22], and within companies responsible for biodiversity offsets [24].

The consideration of ES could improve greatly the societal impact of biodiversity offsets. In some countries, residual direct and indirect impacts of developments projects threaten the survival of local populations due to loss of biodiversity and ES [5]. Some biodiversity offset strategies sometimes negatively affect populations [24]. For example, offsets implemented far from the impacted site (*off-site* offsets [2]) will increase inequalities, as the impacted population will not be compensated in terms of biodiversity or ES provision [4,10]. Moreover, some biodiversity conservation schemes decrease ES access, when implementing a protected area, for example [10,11]. The integration of ES in biodiversity offsets planning could lead to fairer offsets, that consider those kinds of impacts on important ES and livelihoods.

People living in drylands are especially highly dependent on the provision of ES. Indeed, over one-third of the world's population live in drylands [25,26], that are usually vital for the provision of forage for livestock [26]. Other important ES provided are, for example, food production, medicinal plants and fuelwood, water supply (whose availability is limited and variable) for drinking, irrigation and supporting fauna and flora, as well as cultural services related to tourism, spirituality, creation of indigenous knowledge and aesthetics [25–28].

Ecosystem services are therefore essential to people, and their consideration in biodiversity offsets would give a more social, economic and health scope to offsets mechanisms. How could we integrate ES into biodiversity offsets for economic development projects? We are still limited on how diverse ES should be in a specific landscape [22] and on how to prioritise ES [5].Therefore, a systematic identification and prioritisation procedure is needed to ensure that the targeted ES are representative of the area [22].

As ES are attached to beneficiaries, the involvement of stakeholders is vital for identifying bundles of ES to prioritise. The lack of integration of different stakeholders is currently one of the limiting factors for the consideration of ES in biodiversity offsets [11,24], even though it is a recommendation for offsets planning [7]. Stakeholder needs can be assessed through, e.g., questionnaires and individual interviews (e.g., [29–31]), or participatory approaches such as focus groups (e.g., [23,29–32]). However, the perceived ES may vary according to stakeholder categories (e.g., [29]) or gender (e.g., [30]), but these characteristics can be considered through the ES identification process. Identification of ES should also take into account the landscape scale via an ecosystem approach for the design and implementation of biodiversity offsets [7]. Some studies showed already the links between landscape and ES provision (e.g., [30–32]). Identifying the ES provided by the different landscape units should be possible when working in conjunction with local stakeholders.

Our study investigates the prioritisation of ES through a systemic approach across various landscapes. We assumed that different landscape units provide different ES, and that ES are perceived differently depending on stakeholder categories and gender. Our case study takes place in rangelands landscapes in the drylands of southern Kazakhstan, in a context of uranium mining activities. We mobilised the CICES framework for a standardised method, through interviews with a range of local stakeholders. Eventually, our study aims to propose a framework for the identification of ES at a landscape scale, that can be used to integrate ES into the biodiversity offset scheme.

## 2. Materials and Methods

### 2.1. Study Site

The study area was located in the Sozak district of the Turkestan province in Kazakhstan (46°1′18.00″–43°23′44.1″ N and 67°6′10.19″–69°20′31.4″ E). This district comprises the sandy desert of Muyunkum, where the Muyunkum and Tortkuduk uranium mines are located (Figure 1).

The Sozak district is a dry, mid-latitude steppe and mid-latitude desert climate [33]. Mean annual precipitation (MAP) is less than 200 mm/year [34]. According to the Muyunkum Central site project's environmental impact assessment (EIA) carried out in 2011 (hereafter called *project's EIA*), on a more local scale, the climate is continental with temperatures ranging from −30 °C in winter to +40 °C in summer.

Our study site is not limited to only desert, but also comprises a mosaic of landscapes, including:

- The Muyunkum sandy desert, where MAP is 155 mm and during summer and the soil surface temperature can reach 60 °C according to the project's EIA. It is composed of dunes and shrublands, especially of *Haloxylon ammodendron*. (C.A. Mey.) Bunge. [34,35]. Muyunkum is considered as good winter pasture because of the shrubby vegetation that can be found, even after heavy snowfalls [34].
- The steppe to the south and west of the sandy desert. Steppes are grazing or grasslands areas are found, that allow livestock farming and wheat cultivation [33].
- On the northern of the sandy desert, the presence of the Shu River, which forms in Kyrgyzstan, has created riparian and flooded zones where reed (*Phragmites* spp.) is the dominant species. Reeds are consumed by cattle but also cut and stored as winter fodder by villagers living along the river [35,36]. The riparian zone includes *Tamarix* spp., and the herbaceous species *Agrophyron* spp., *Festucca* spp. and *Artemisia* spp. are present in the seasonally flooded zones [36].

- The Betpak-Dala steppe that lies north of the Shu river. This steppe is described as a clay desert, comprising sparse vegetation that incudes *Artemisia* spp. and *Salsola* spp. and several annual species, representing an important and rich source of protein for herbivores in the early spring [34–36].
- An area of salty, clay desert between the Shu river and the Muyunkum sandy desert, where several halophyte plant species are present [35].
- The Karatau mountains in the south-west, where the majority of precipitation falls, according to the project's EIA.
- A salty lake area in the south-east, comprising the Kyzylkol lake and sacred pond of the religious site of Baba Tukti Shashty Aziz mausoleum, on both sides of the border of the province of Djambul.

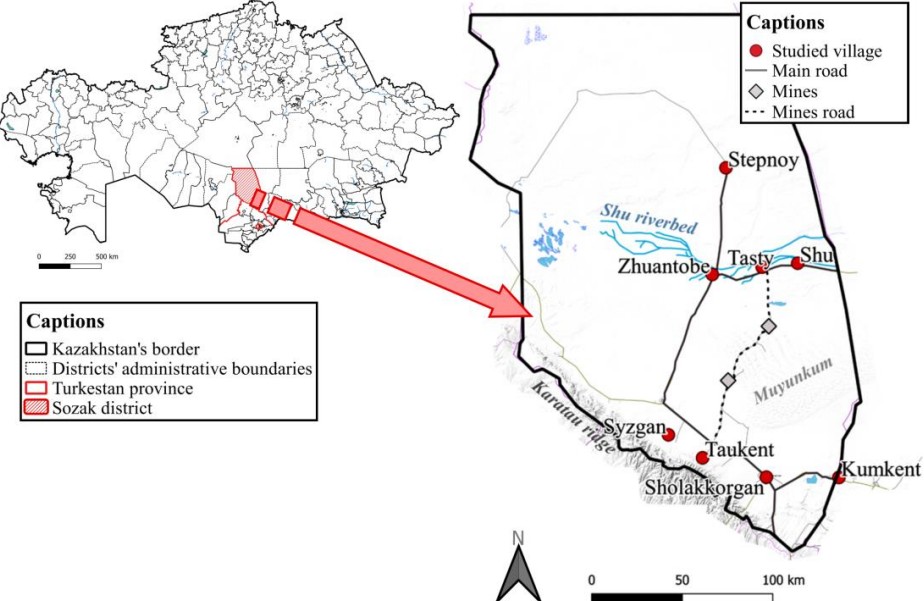

**Figure 1.** Location of Sozak district in Kazakhstan. Ecosystem services were identified in and around eight villages across the landscape.

## 2.2. Main Economic Activities: Livestock Farming and Uranium Mining

The population of the Turkestan province is 2.7 million people, resulting in a population density of 23 people/km², and the population of the Sozak district is 62,000 people, resulting in a population density of 1.5 people/km² [37]. According to the project's EIA, livestock farming is the primary activity in the study area, followed by crop production.

The environment of Kazakhstan is favourable to mobile pastoralism [38]: the rangelands represent 60% of the country, i.e., 189 million hectares [36]. Agriculture, including livestock farming, contributed 4.4% of the country's GDP and accounted for 18% of employment in Kazakhstan in 2017 [39]. Turkestan province is one of the most important regions for livestock farming, with the larger concentration of small-scale farms in 2016 and the highest number of registered agricultural cooperatives and cooperative members in 2018 [39]. Nevertheless, over the past 150 years, nomadic pastoralism has declined and now accounts for very few herders [34]. The use of rangelands for livestock production has undergone major changes, from state-owned farms in the Soviet era, to the development of private livestock systems since the mid-1990s [34,35,40]. Nowadays, the richest herders with large herds can exclusively rent grazing areas with access to wells, via a semi-privatization mechanism [35]. Herders with less livestock tend to keep animals in the proximity of villages [35,40]. Herders can grow and store hay for use in the winter months, mostly as fodder for dairy cows and calves [40,41]. Small livestock (sheep and goats) and large livestock (cows, horses and camels) are raised for different animal products

(dairy, meat and wool) [38,41]. In 2009, Turkestan province had 3,415,000 sheep and goats, 716,000 cows and 144,000 horses (no data on camels) [41].

Kazakhstan's subsoil is rich in various mineral resources including oil, gas, uranium, coal, copper, zinc, gold, chromium, manganese, iron and lead [42,43]. In 2016, the extractive sector contributed almost 30% of the country's GDP [44]. The Shu-Syrdarya mining region, in which our study area is located, was discovered and explored between 1971 and 1991, and is now the largest uranium mining region in Kazakhstan [45]. The Kazakh–French joint venture KATCO (Kazatomprom–Orano) owns the mines at the Muyunkum and Tortkuduk deposits in the Muyunkum sandy desert of the Sozak district. Since the 1990s, the in-situ recovery (ISR) technique became widespread for uranium extraction, used for ore located between impermeable soil layers [37]. Compared to older mining methods, this technique has less surface damage, no waste rock or tailings storage and lower remediation costs [45,46]. Nevertheless, there are risks of local underground contamination due to leaching reagents and metals in solutions [46]. In addition, the implementation of mining projects can also have impacts on ES, such as livestock-related ES, by decreasing access to grazing areas, for example.

### 2.3. Identification of Ecosystem Services to Include in Biodiversity Offsets

Figure 2 summarises the methodology applied. Each methodological point is described in more detail in the following sub-section, with application in our study area.

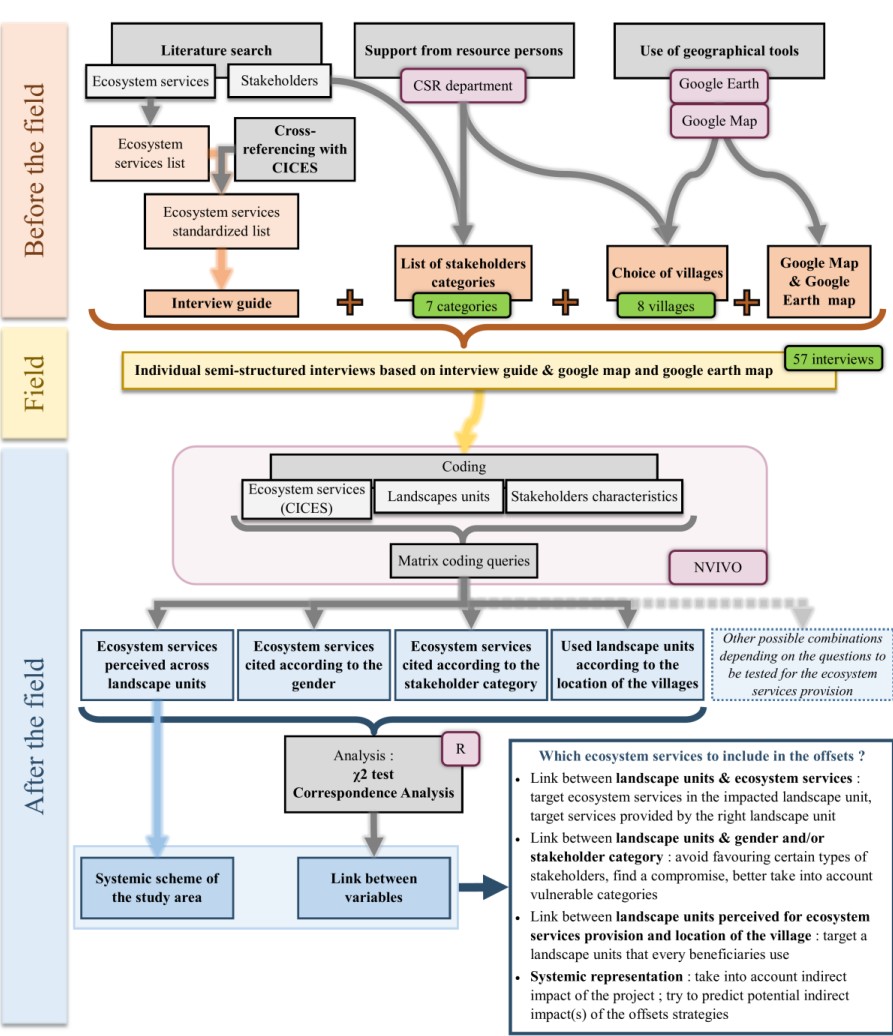

**Figure 2.** Methodology applied for the identification of ecosystem services to be included in biodiversity offset strategies (green: case study of the Sozak district in Kazakhstan; purple: software and resources used; CSR = corporate social responsibility).

### 2.3.1. Construction of the Interview Guide

To identify the ES throughout the landscape in the Sozak region, we carried out a literature search and developed a questionnaire before then interviewing local stakeholders. We performed a literature review on the ES provided by pastures in Kazakhstan, Central Asia and drylands. The keywords used for the bibliographic research were: 'ecosystem + services' and: 'Kazakhstan' or 'Central Asia' or 'drylands' or 'arid + lands' or 'pastures' or 'rangelands' or 'grasslands', leading to 11 studies, reports, papers and book chapters [25–28,47–53]. We then compiled the ES and cross-referenced them with the CICES [16] to produce a standardized list of ES. This list was the basis for our interviews with stakeholders. We kept the CICES class level for some ES and remained at the CICES group level for other when the CICES division level was too vague to guide our questions, and the CICES class level too specific (e.g., the Biomass division includes the Wild Plants for food, materials or energy group, which includes the classes Wild plants for food, Wild plants for energy, etc. [16]). The interview guide, which is based on this list, is available in Table S1.

### 2.3.2. Selection of Local Stakeholders and Villages in Sozak District

Although uranium mines are located in the sandy desert of Muyunkum, this area is not inhabited by a permanent population and is only used as a winter pasture. The majority of the local population live along the Shu river and the foothills of the Karatau mountains. Therefore, we selected stakeholders from eight villages located around Muyunkum: Taukent, Syzgan, Sholakkorgan and Kumkent in the south and Zhuantobe, Tasty, Shu and Stepnoy in the north (Figure 1). Among these eight villages, four (Taukent, Sholakkorgan, Tasty and Shu) work closely with the French–Kazakh Joint-Venture KATCO. KATCO finances infrastructure such as schools, or gifts coal to the inhabitants. We chose stakeholders based on: categories of stakeholders (Table 1) and a balanced gender dimension. Stakeholder categories were determined through a stakeholder mapping report (performed by Central Asia agency for KATCO in 2019), discussions with KATCO Corporate Social Responsibility (CSR) department, and studies about Kazakh livestock systems and management [36,40,54] (Table 1). They represent the different types of local stakeholders in the study area.

**Table 1.** Stakeholder categories description, number and proportion of participants per category.

| Category | Description | Number of Participants | Percentage of Participant (%) |
|---|---|---|---|
| Local authority or his deputy | Akim or his deputy. It is at the level of the Akimat that decisions on land planning are made. It is also at this level that the grievances and various demands of the inhabitants are received | 10 | 18 |
| Elder | These are older men and women. They are respected and may have knowledge of current land and natural resource use, but also of the past. Among them, there are 'veterans', elderly men with a special status: organized as councils, they are asked by inhabitants for their opinion on certain issues and they can act in some decision-making processes related to the life of the village | 8 | 14 |

**Table 1.** *Cont.*

| Category | Description | Number of Participants | Percentage of Participant (%) |
|---|---|---|---|
| Herder farmer | Herders and farmers in cooperatives: in farms, often a family business, with a big herd. They move their livestock every season. They are key actors in the ecosystem services provided by the different kinds of pastures; but also, stakeholders not necessarily organized in business but with a big herd and willing for production, and whose herd move pasture every season. | 7 | 12 |
| Social and health worker | Health workers are nurses or doctors, for example. People working in the social field deal with isolated people, large families, disabled people, elderly and/or sick people, either through providing legal and financial support or social support. This category can account for villagers' health problems that may be linked to environmental problems, as well as ecosystem services that are important to vulnerable individuals or families. | 7 | 12 |
| Mother with many children | Women with at least six children. They can receive a medal, of different levels depending on the number of children. They are among the categories eligible for social and financial assistance, often in the lists of vulnerable persons. They may have different perceptions and/or needs for ecosystem services for their family. | 7 | 12 |
| Teacher | Teachers of different levels, from school to high school and in different fields. They are educated people who may have other types of knowledge. | 8 | 14 |
| Inhabitant | A random person, regardless of status and occupation: unemployed, driver, veterinarian, shop owner, media, etc. | 10 | 18 |
| Total | | 57 | 100 |

### 2.3.3. Interviews in Sozak District

Between 28 June 2021 and 17 July 2021, we conducted a total of 57 individual interviews (Table 1). In each of the villages, seven people were interviewed (except for Sholakkorgan where we interviewed eight people). The first person we met in each village was usually the *Akim* (mayor), whose office had identified the interviewees according to the characteristics we were looking for. Interviews were conducted at the *Akimat* (town hall) of each village and were translated simultaneously by an interpreter. We interviewed 34 men and only 23 women, because *Akims* and herder farmers were always men. In total, 86% of the 57 participants were breeders of livestock, some based in agricultural cooperatives (herder farmer in Table 1) and others who were small-scale livestock farmers.

The questionnaire was comprised of boxes, with each box corresponding to a CICES group or class of ES (Table S1). Our questions were not fixed and were semi-structured. For example: 'Are crops grown in the area?' and depending on the answer we then asked: 'For what purpose?' 'Where are they grown?' and so on. The sub-questions, which sought to identify specific details about each ES, depended on the individual we met, their answers and perceptions. For example, we were able to go into detail about the soil quality regulation services with an individual X, but went to another level of detail on the groundwater services with individual Y.

Several studies have shown that the landscapes perceived by local beneficiaries may be different from those identified by academics, e.g., a unit identified by academics may actually be two units for local stakeholders (e.g., [32]). In addition, the use of maps makes

it easier to introduce the subject of ES provision [30]. Thus, we tried as far as possible to locate the spot(s) and/or landscape(s) where the ES occurred using printed maps from Google Earth [55] and Google maps [56] (up to a radius of 20 km around the village of the interviewee), or orally, if the interviewee could not locate it on a map.

During the interview, all CICES groups of ES were discussed, even though some ES may not be directly relevant for biodiversity offset mechanisms. In addition, we were not limited to our list of CICES groups: some ES were added when required, as stated by the participant as the discussion progressed. Thereby, we ensured that the main benefits provided by services were not overlooked [22].

### 2.3.4. Data Analysis

Interview data were processed using NVivo [57], a qualitative data analysis software. Interviews were described according to interviewees' attributes: gender, stakeholder category and village. Then, interviews were coded by ES and landscape units. For ES codes, we used the hierarchical structure of the CICES. From the CICES class level, details on the ES and the benefits from the ES were added. Where necessary, ES categories were added, following CICES recommendations [16]. The landscape unit codes were added according to the landscape described for the provision of ES (maps from Google Earth and Google maps or oral location). However, some ES had no landscape boundaries, such as services related to existence value [22] ('Characteristics or features of living systems that have an existence value' in CICES [16]). In these cases, we coded certain ES in a special category that we called *no landscape frontier*.

From the organisation and coding of data, we could create matrix coding queries with the NVivo software. We used matrix coding queries to analyse in detail the relationships between the following sets of data, and to produce tables of data that included the following information for each cell (x,y):

(i) Ecosystem services (each level of CICES) and landscape units: number of interviewees who cited the ES (x) in the landscape unit (y). Subsequent analysis based on this matrix will verify the hypothesis that different landscapes provide different ES.

(ii) Stakeholder categories and ecosystem services (each level of CICES): number of interviewees from a category (y) who cited the ES (x).

(iii) Gender and ecosystem services (each level of CICES): number of male and female interviewees (y) who cited the ES (x).

Further analysis based on matrix (ii) and (iii) will test the hypothesis that stakeholders perceive ES according to their category and their gender.

(iv) Location of the villages and landscape units: number of interviewees from a village (y) who cited the landscape unit (x). In the case where ES are linked to the landscape units (analysis based on matrix (i)) but the stakeholders' categories and gender (analysis based on matrix (ii) and (iii)) do not guide the perception of ES, we tried to find out what did. Here, we test the location of villages as a criterion.

Before the following analysis, we removed the CICES ES class if it was cited by less than 10% of the participants (i.e., by five or fewer participants).

A $\chi^2$ test was then performed on these data using the software R [58] to check the independence of variables. If the H0 hypothesis (independence between the two variables) was rejected (meaning that the obtained *p*-value < 0.05), and our variables were significantly related, then we performed a correspondence analysis (CA) to visualise the nature of the relationships. Instead of showing all CA, for the purpose of readability, only the positive relationships according to the CA are given and illustrated by ES section (provisioning, regulation and maintenance or cultural) for all level of CICES according to its hierarchical structure. The positive relationships are based on the respective contribution of variables to CA dimensions (axes).

As an overall result, we show in a systemic way the perception of ES provided by the different landscape units, depending on whether they are provisioning, regulation and maintenance or cultural services. An ES is shown with a landscape unit only if it was cited

by at least 10% of the participants as being present at that unit, even if it was at first selected for analysis because it had been cited by 10% of participants in total (i.e., with all landscape units combined).

## 3. Results

From our literature study, we found a total of 116 mentions of ES. When classed into the CICES system, we organised ES into 10 divisions, 22 groups and 46 classes of ES among the 15 divisions, 36 groups and 90 classes initially present in the CICES. We also added two services; *cultivated plants for fodder* and *wild plants grazed by reared animals*, following the structure of the CICES. These services are not final ES, according to the cascade model [13,17], but are important to take into account since livestock farming and production are major activities in our study area.

From our interviews with stakeholders, 300 ES in total were identified by interviewees, which we grouped into the 61 classes of CICES. By keeping the classes of ES in which 10% participants (i.e., at least six participants), mentioned ES, we obtained 37 CICES classes of ES of interest (Table 2).

**Table 2.** Number of perceived ecosystem services and CICES class before and after the 10% selection.

| Ecosystem Service Section | Number of Perceived Ecosystem Services | Number of Ecosystem Service Classes (CICES) | Selected Ecosystem Services Classes (CICES) (>10%) |
|---|---|---|---|
| Provisioning (biotic) | 141 | 14 | 11 |
| Provisioning (abiotic) | 20 | 14 | 8 |
| Regulation and maintenance (biotic) | 71 | 14 | 6 |
| Regulation and maintenance (abiotic) | 12 | 6 | 3 |
| Cultural (biotic) | 41 | 10 | 5 |
| Cultural (abiotic) | 15 | 3 | 3 |
| **Total** | 300 | 61 | 36 |

By gathering 300 perceived ES into 37 classes of ES, i.e., almost 10 times less, further analysis could be simplified. Provisioning services are the most perceived (total of 161 ES in 19 classes of CICES) than regulation and maintenance (total of 83 ES and 20 classes) and then cultural services (total of 56 ES and 13 classes).

Moreover, participants were generally able to describe the provision of ES across various landscape units. Figure 3 illustrates the 7 landscapes units described by interviewees. In addition, the participants also located ES in a landscape unit termed *village*. The salty clay steppe area between the Shu river and the Muyunkum sandy desert was the only landscape unit not mentioned by stakeholders (see Section 2.1 in Materials and Methods).

In the further results, we keep the terms used by stakeholders to describe the landscape units. The corresponding landscapes (see Section 2.1 in Materials and Methods) are:

- The Muyunkum sandy desert: called *Sand* by participants
- The steppes: called *Steppe*
- The Shu river and riverbanks: called *River*
- The steppes of Betpak-Dala: called *Betpak-Dala*
- The Karatau mountains and its foothills: called *Mountains* and *Foothills*
- The salty lakes area: called *Lakes*

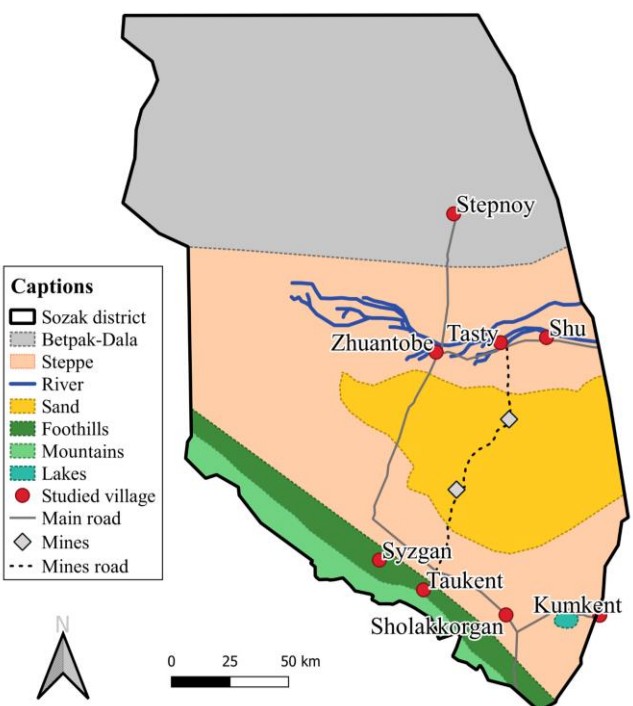

**Figure 3.** Landscape units as described by interviewees in the context of ecosystem services provision. Main road, Mines and Mines road are not landscape units but supplementary information.

### 3.1. Provision of Ecosystem Services through Landscape Units

All $\chi^2$ tests rejected the hypothesis H0 of independence between the landscape unit and ecosystem services' variable ($p < 0.05$) both (i) at each level of the CICES (from least accurate to most accurate) and (ii) within each section (provisioning, regulation and maintenance, cultural) (Table S2). Therefore, the distribution of perceived ES provision across landscape units and the differences attributed was not random. Our first hypothesis is verified, meaning that some ES are linked to specific landscape units.

Several correspondence analysis (CA) were performed to visualize this link. The CA between ES section (provisioning, regulation and maintenance, cultural) and landscape unit is provided as an example (Figure 4). Positive links between ES section and landscape units in this CA are explained in Table 3. From this example, we observed that provisioning services are mostly provided by the steppe, the sand and the village for the biotic section (the provisioning biotic section includes, for example, cultivated plant, wild plants grazed by reared animals, and reared animals) and by the foothills and Betpak-Dala for the abiotic part (e.g., surface water or mineral substance). Cultural services are linked to the no landscape frontier category (e.g., existence or bequest value), to the lakes (e.g., spiritual, symbolic and other interactions) and to the river units (e.g., activities promoting health, recuperation, enjoyment). Biotic regulation and maintenance services (e.g., maintaining nursery population and habitats) are mostly linked to no landscape frontier category, whereas abiotic services do not seem to be link to specific landscapes units. The landscapes unit mountains does not provide specific ES at this level of CICES.

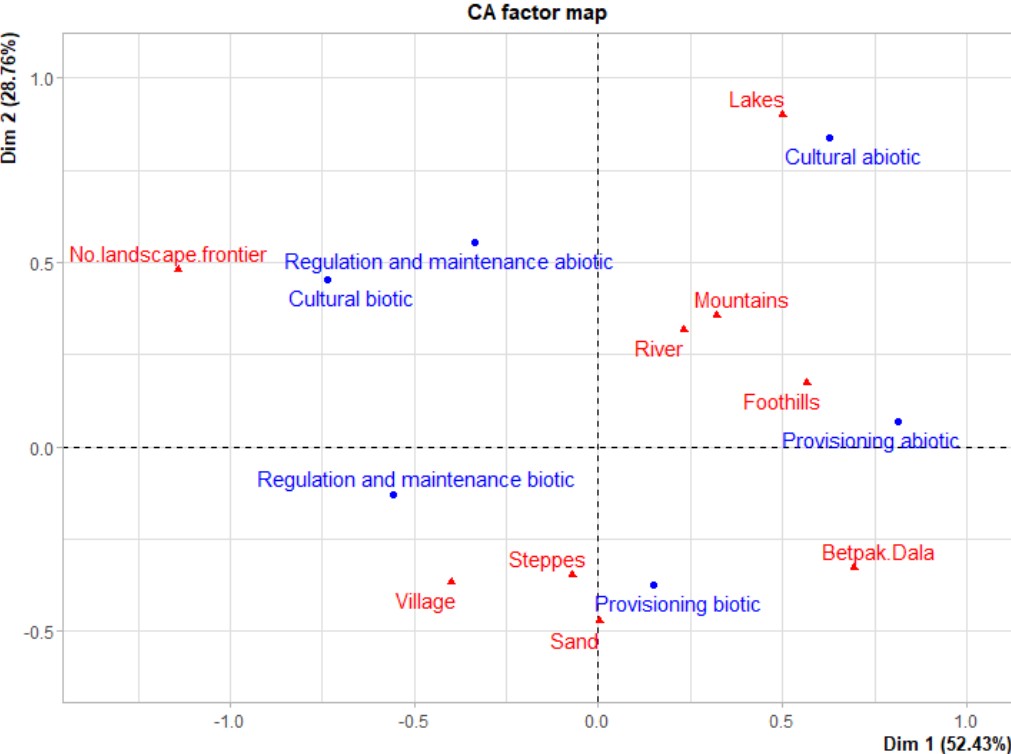

**Figure 4.** Example of correspondence analysis (CA) factor map. Here is the CA performed on ecosystem services CICES section and landscape units. In blue: ecosystem services CICES section, in red: landscape units.

**Table 3.** Links between ecosystem services CICES section and landscape units, according to the respective contributions of each variable to dimensions (axes) 1 to 3 (3 axes remained) and their relative position on the contributed axes in the correspondence analysis.

| Dimension | Ecosystem Service Section | Landscape Unit |
|---|---|---|
| 2 | Provisioning biotic | Steppe, Sandy area, Village |
| 1 and 3 | Provisioning abiotic | Foothills, Betpak-Dala |
| 1 | Regulation and maintenance biotic | No landscape frontier |
| / | Regulation and maintenance abiotic | ? |
| 1 | Cultural biotic | No landscape frontier |
| 2 and 3 | Cultural abiotic | Lakes, River, No landscape frontier |

The CA were performed on each of the other CICES levels: division, group, class by service section: provisioning, regulation and maintenance, cultural, resulting in a total of nine additional CA. Positive links between ES and landscape units are illustrated in Figures 5–7 and described in Table S3. We have followed the CICES hierarchical structure for clarity in the illustration. The legend, i.e., to which landscape unit the letters correspond, is available in the figure titles. When a landscape unit box is on an ES box, then this landscape units is more linked than the others to this ES.

The $\chi^2$ test, followed by CA on each level of CICES, and schematized in Figures 5–7, verified and illustrated our hypothesis that different landscapes units provide different ES. Few of the CICES classes of ES were not linked to a specific landscape unit, as wild plants for energy, reared animals for materials (health), wild animals for nutrition, and surface water and groundwater for watering livestock (Figure 5), as well as wind protection, pollination, regulation of temperature and humidity and seed dispersal (Figure 6). Therefore, 75% of the class of ES were provided by specific landscape unit(s) according to the interviewees.

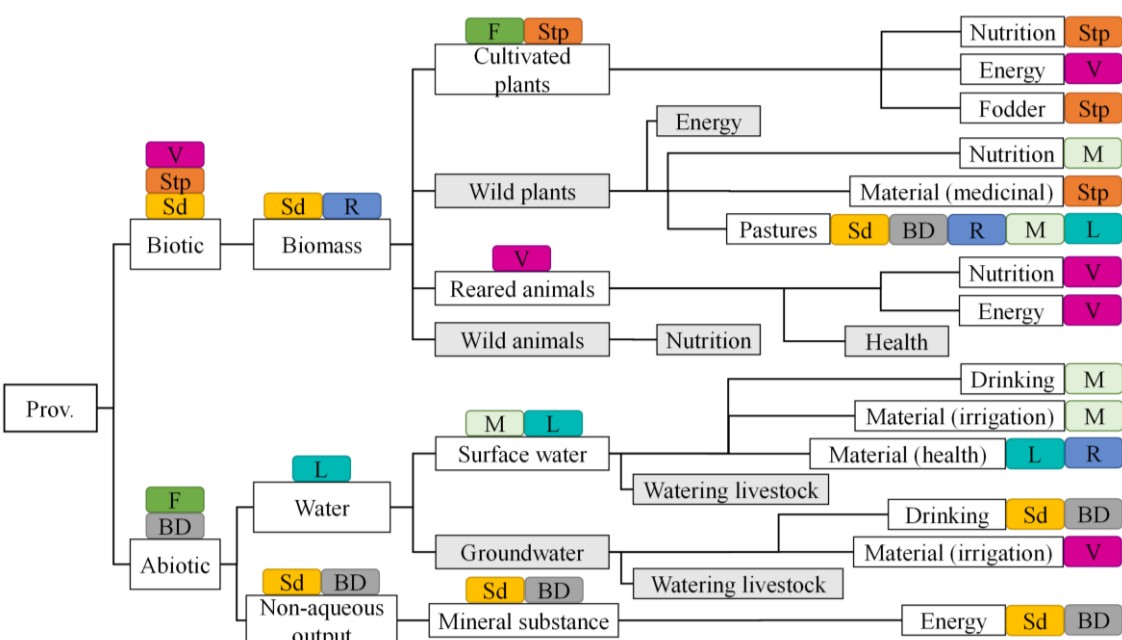

**Figure 5.** Hierarchical diagram following CICES representing the links between provisioning (Prov.) ecosystem services (ES) and landscape units according to their respective contribution to CA dimensions. When a landscape unit box (in colour) is on an ES box, then this landscape unit is more linked than the others to this ES (Stp = Steppe; Sd = Sand; V = village; M = Mountains; F = Foothills; L = Lakes; R = River; BD = Betpak-Dala). When an ES box is grey, it is not linked to any landscape unit according to its contribution to CA dimensions.

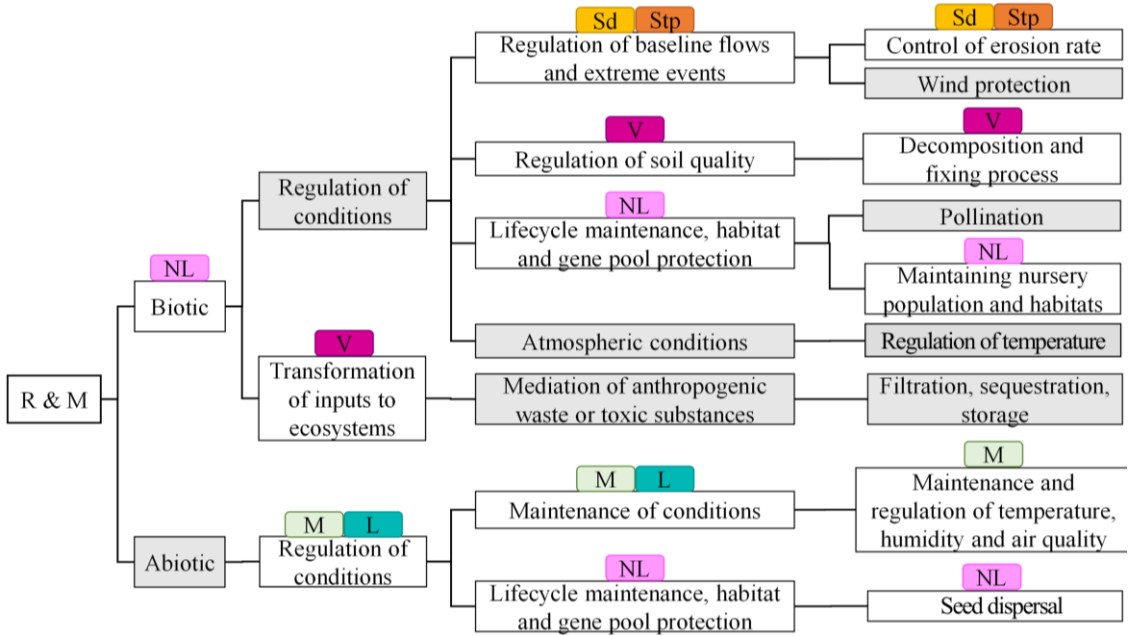

**Figure 6.** Hierarchical diagram following CICES representing the links between regulation and maintenance (R&M) ecosystem services (ES) and landscape units according to their respective contribution to CA dimensions. When a landscape unit box (in colour) is on an ES box, then this landscape unit is more linked than the others to this ES (Stp = Steppe; Sd = Sand; V = village; M = Mountains; F = Foothills; L = Lakes; R = River; BD = Betpak-Dala; NL = No Landscape Frontier). When an ES box is grey, it is not linked to any landscape unit according to its contribution to CA dimensions.

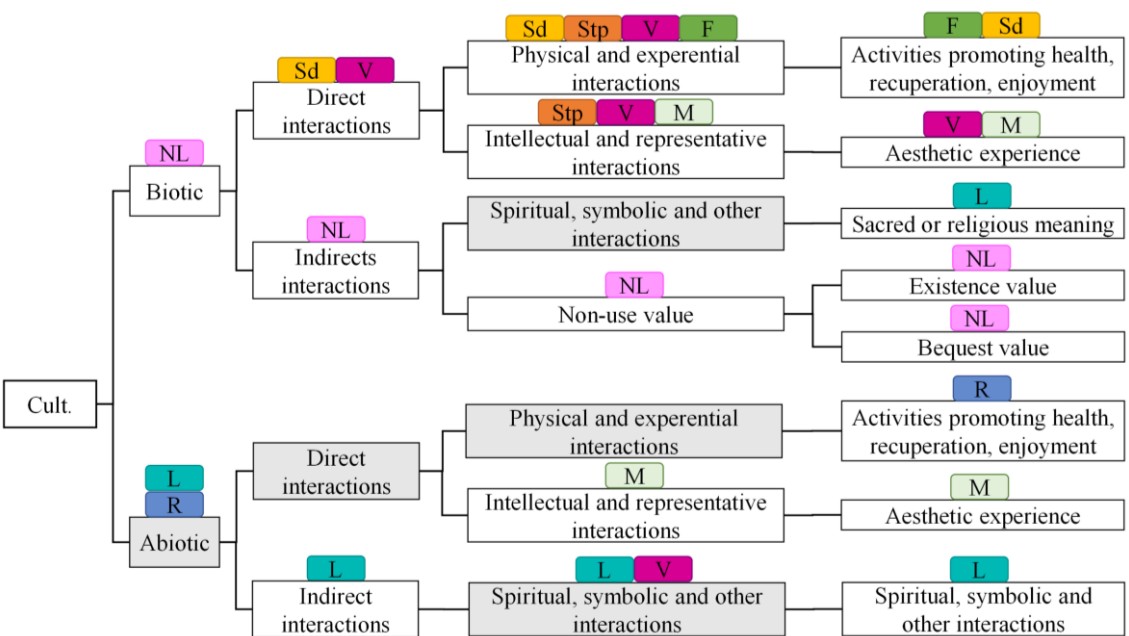

**Figure 7.** Hierarchical diagram following CICES structure representing the links between cultural (Cult.) ecosystem services (ES) and landscape units according to their respective contribution to CA dimensions. When a landscape unit box (in color) is on an ES box, then this landscape unit is more linked than the others to this ES (Stp = Steppe; Sd = Sand; V = village; M = Mountains; F = Foothills; L = Lakes; R = River; BD = Betpak-Dala; NL = No landscape frontier). When an ES box is grey, it is not linked to any landscape unit according to its contribution to CA dimensions.

### 3.2. Preference of Ecosystem Services according to Stakeholder Category and Gender

No $\chi^2$ test rejected the hypothesis H0 of independence ($p > 0.05$), no matter the level of CICES. Thus, there was no significant link between stakeholder categories and ES and between gender and ES. Our second hypothesis is rejected, meaning that stakeholders do not perceive ES according to their category or gender. In the case of Sozak district, other criterion(s) guides the perception of ES.

### 3.3. Links between Stakeholder Village Location, Landscape Unit and Ecosystem Services

The $\chi^2$ test rejected the hypothesis of independence between the village location and landscape unit variables ($p = 2.005 \times 10^{-11} < 0.05$). Therefore, the described landscape units in terms of ES provision are not random and is linked to the location of the interviewees' village.

CA were performed to visualize this link (Figure 8). There was also a contrast between villages in the north and south of the study area. Participants tended to identify more frequently the landscape units they lived close to in terms of ES provision. Interviewees in the south (from Kumkent, Taukent, Sholakkorgan and Syzgan villages) tended to perceive the mountains, foothills and lakes as providing most of the services, while those in the north (Zhuantobe, Tasty, Shu and Stepnoy villages) tended to perceive the river and Betpak-Dala as providing most of the ES. The landscape units no landscape frontier, village, steppe and sand were common to all interviewees, in the context of ES provision. This result showed that stakeholders described the landscape units for ES provision according to the location of their village within the landscapes, rather than the stakeholder category or gender.

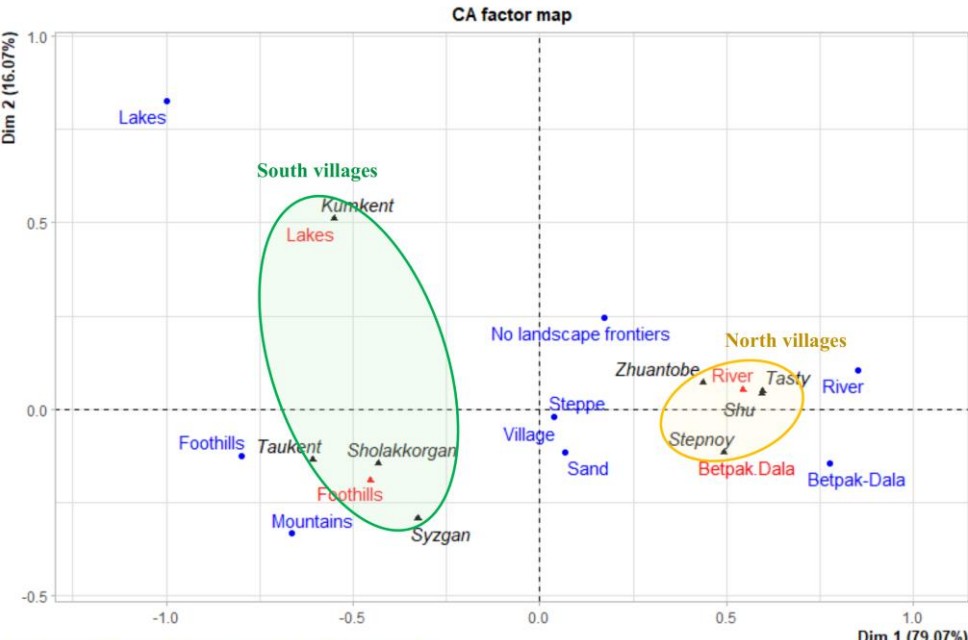

● Landscape units for ecosystem services provision
▲ Landscape units where are located the villages
▲ Villages

**Figure 8.** CA performed on participant's village location (in red) and landscape units described for ES provision (in blue). In black: name of the villages as supplementary variable. For blue variables, the unit Foothills comprises Taukent, Sholakkorgan and Syzgan; the unit Lakes comprises the village of Kumkent; the unit River comprises Zhuantobe, Tasty and Shu; and Betpak-Dala comprises the village of Stepnoy. The CA factor map shows a contrast between villages in the south (green ellipse), and those in the north (yellow ellipse).

*3.4. Systemic Representation of the Perception of ES Provided by the Different Landscape Units*

From the results above, we considered the inter-relation between ES throughout the landscape (Figures 9–11). Not all ES are shown, as some ES were cited by 10% of the total number of participants, but not in all landscape units (e.g., mineral substances for energy were cited by 15% of participants in total, but only by 2% in the steppe, 7% in sand and 7% in Betpak-Dala). In addition, services related to reared animals were linked to species: if less than 10% of participants cited a species for a given service, it was not represented (e.g., sheep manure is used by 22% of participants and so was represented, but horse or cow manure was only used by 2% and so was not shown). A detailed description of stakeholders' perceptions of ES is provided below.

3.4.1. Provisioning Services

All landscape units except the lakes were identified by at least 10% of participants as pasture (Figure 9). The steppe is an important grazing area (85% of participants), in all seasons. The sandy area is an important winter grazing area (42% of participants). Betpak-Dala and the mountains are more of a summer grazing area. The steppe and foothills allow for the cultivation of winter fodder (Figure 9) (42% and 21% of participants, respectively), mainly clover and corn. Other strategies are implemented for winter fodder storage (Figure 9) via cutting and storing wild plants: around the Shu River, people store reeds (17% of participants). Steppe grasses can also be stored as hay (26% of participants). The reared animals are important contributors to the nutrition of the district's population (Figure 9). Cows and sheep are the main contributors: 77% of the participants consume dairy products from cows, 28% beef, and 66% mutton.

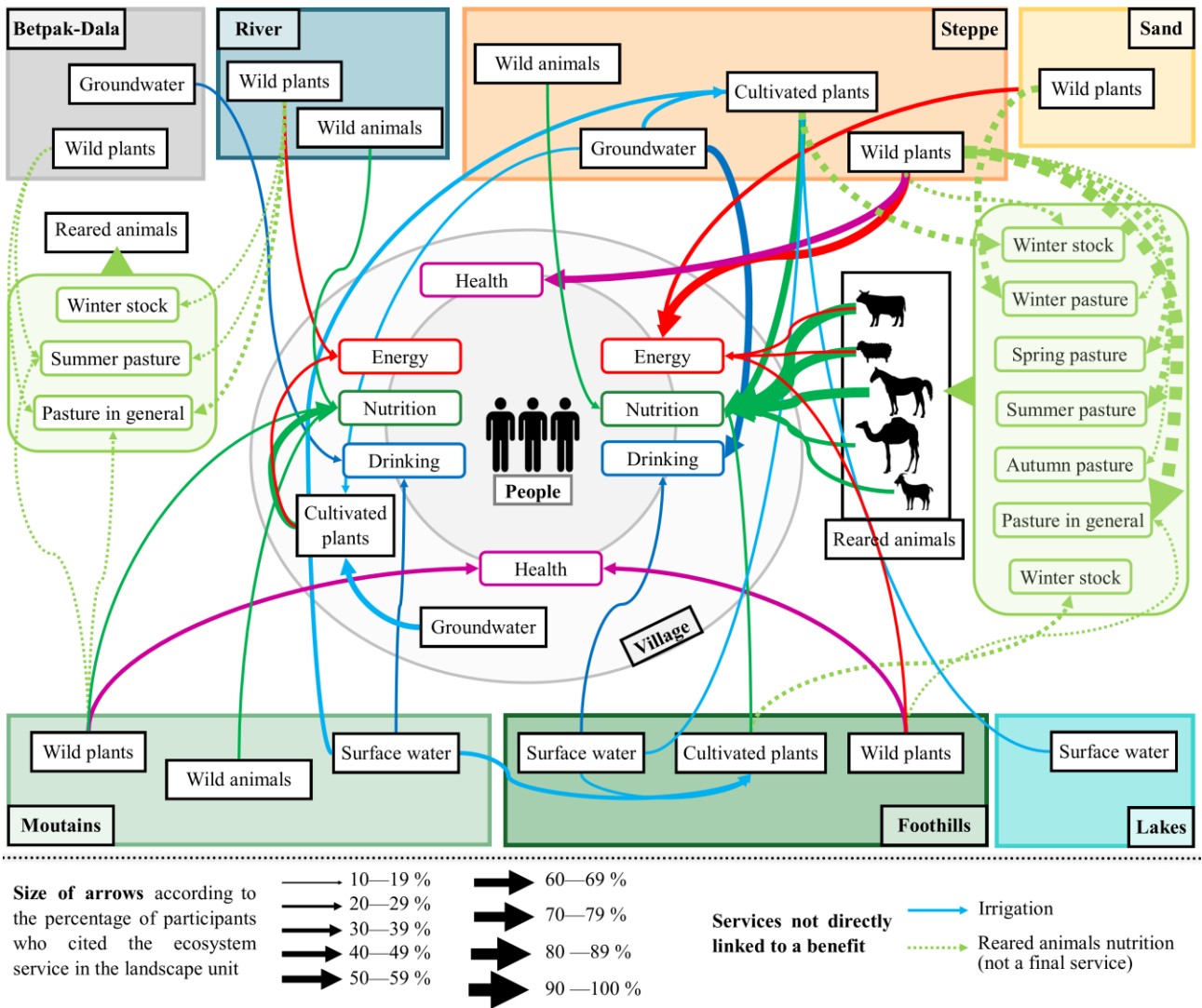

**Figure 9.** Systemic scheme of provisioning services provision across the landscape units.

The other sources of food are from cultivated plants (Figure 9) in the steppe (mentioned by 49% of the participants), and in the foothills (17%): mainly cultivated melons, watermelons, vegetables, as well as some fruit trees. Additionally, 45% of the participants had a vegetable garden and/or a yard with fruit trees, which highly contribute to the food of the families (Figure 9). Finally, wild plants are used for nutrition; they are mostly collected for this purpose in the mountains (19%), including herbs and wild fruits (Figure 9).

The sources of energy from biomass are wild plants collected mainly in the steppe (57%) and the sandy zone (22%) and include mainly woody plants, e.g., *Bayalich* (*Salsola arbuscula* Pall.), *Djingil* (*Tamarix* spp.) in the steppe and Saxaul (*Haloxylon* spp.) in the *sand* that are used, purchased with permission from the authorities or sometimes cut illegally (Figure 9). Sheep and cow manure are also used as energy sources (by 17% and 12% of participants, respectively) (Figure 9). Moreover, some participants use the fallen and dried branches of the village's ornamental trees (15%) (Figure 9).

Wild plants are also used as medicinal or purifying plants and come mainly from the steppe (49%), mountains and foothills (24% and 26%); the species mostly collected is *Adrespan* (*Peganum harmala* L.), used to purify houses (40% of the participants collect it in the steppe, for example, Figure 9).

Drinking water is either obtained from surface water in the mountains and foothills, or from groundwater in the steppe and Betpak-Dala. These water resources are also used for the irrigation of cultivated plants (Figure 9). For surface water irrigation: mountains

contribute 28%, followed by foothills and lakes, according to the participants. For groundwater for irrigation, only the steppe was cited by at least 10% of participants. In the village, families with gardens and yards generally use water from their own well (36%) (Figure 9).

### 3.4.2. Regulation and Maintenance Services

The air quality service is provided by the village trees (31% of participants), which purify the air and filter and stop the dust brought by strong winds (Figure 10). Among participants, 36% found the air quality to be good, compared to 9% who did not. The structure of the Karatau mountain also contributes to the air quality (12%), as a physical barrier (Figure 10). The trees in the village also contribute to the local regulation of temperature and humidity (28%): they make the local climate more pleasant by providing shade and cool air. The mountains also contribute to this, again due to their structure and altitude (12%) (Figure 10). Among the participants, 19% did not perceive this service, especially in the north towards the Shu river and Betpak-Dala.

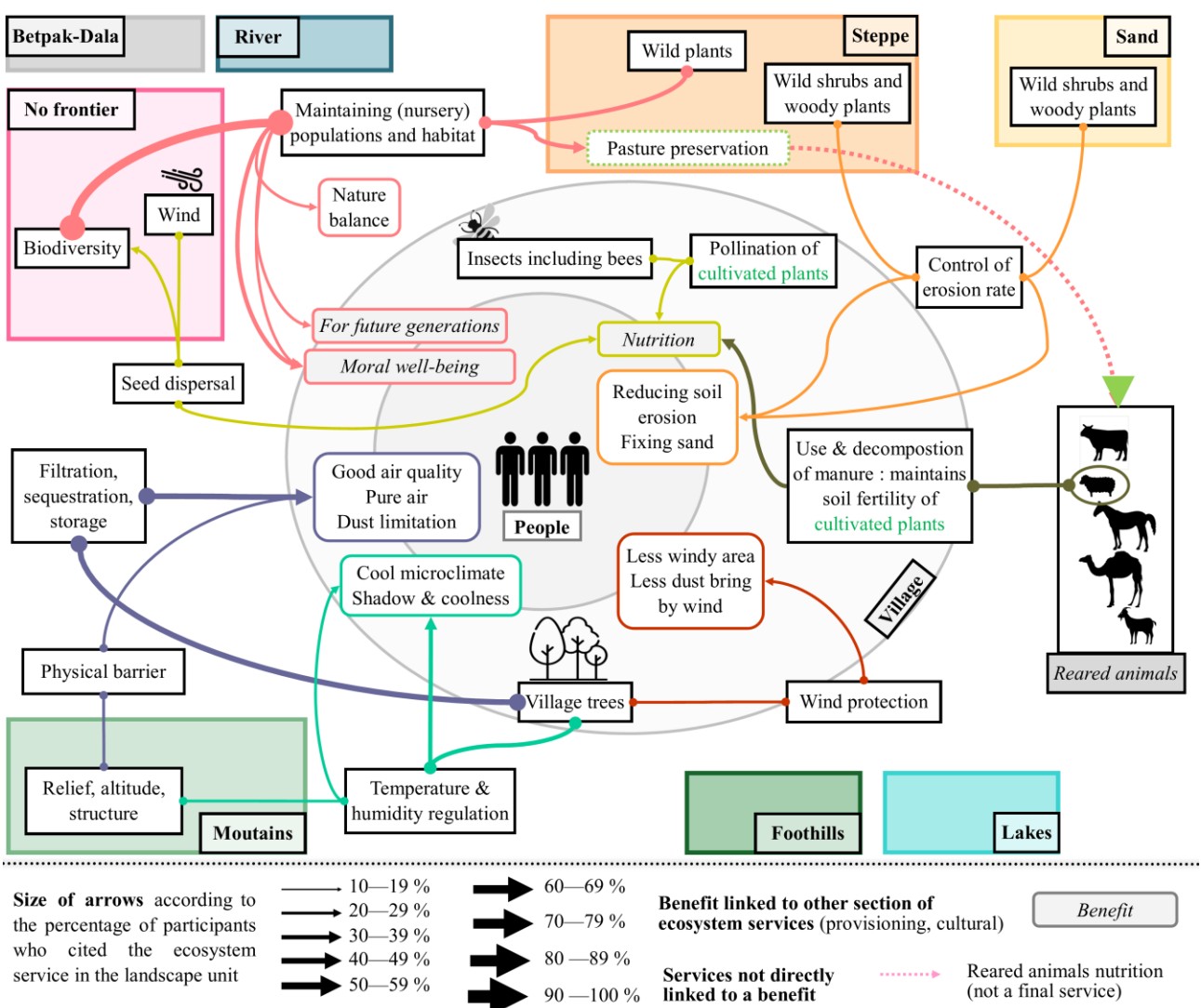

**Figure 10.** Systemic scheme of regulation and maintenance services provision across the landscape units.

With regard to the soil quality service, sheep manure is used by 22% of the participants (Figure 10) for individual vegetable gardens and thus contributes to the nutrition benefit (Figure 9). Wild woody plants Saxaul (*Haloxylon* spp.) contribute to the control of erosion by fixing the soil, mainly in the steppe (17%) and in the sand (19%) (Figure 10).

Regarding pollination and seed dispersal, they were, respectively, enabled by insect pollinators (perceived by 10% of participants for pollination in the village) and wind (perceived as a disseminator by 18% of participants) (Figure 10). Wind was not attached to any landscape unit, as 'the wind is everywhere'.

Finally, biodiversity, which includes wild flora and fauna, is considered important to maintain and/or protect by 71% of participants, and 66% want biodiversity protected to maintain other services and benefits that cannot be included in landscape boundaries (Figure 10). The majority of the participants wish to maintain it and give it importance for cultural reasons: for future generations (17%), but also because fauna and flora are components of nature, which have as much right as humans to exist (21%). Thus, this ES contributes to the maintenance of cultural services (Figure 11). Additionally, the maintenance of local flora, especially in the steppe, is considered important because it is a source of food for reared animals (17% of participants) (Figure 10).

### 3.4.3. Cultural Services

The lakes (10%), mountain (19%), foothills (21%) and the Shu river (24%) participate in the rest and enjoyment of the inhabitants, through direct interactions with their abiotic components (Figure 11). Those interactions result in visits to these landscape units, with swimming, resting and picnics, often with the family. The foothills also participate through biotic components (12%) (Figure 11): for example, through picnics and resting in private gardens, and recreational fishing.

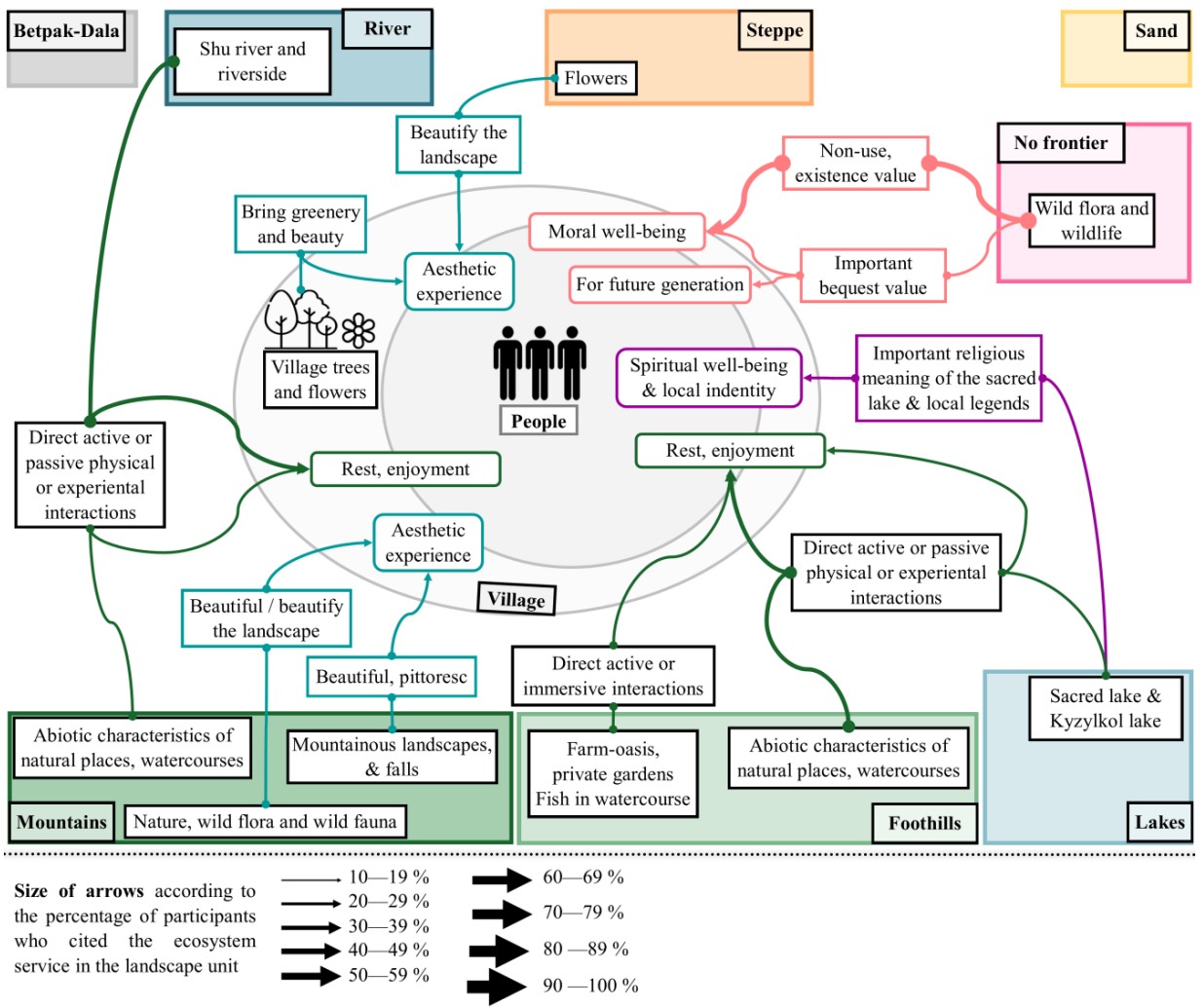

**Figure 11.** Systemic scheme of cultural services provision across the landscape units.

The mountainous landscape with its waterfalls (14%), but also its fauna and flora (10%), is appreciated for its beauty (Figure 11). The flowers of the steppe that bloom in spring (10%) are also appreciated for their colour (described as a 'multicolor carpet') (Figure 11). The flowers and ornamental trees in the villages are also important for the inhabitants, because they make the village more beautiful and provide it with greenery (19%) (Figure 11).

The lake of Baba Tukti Shashty Aziz mausoleum is sacred and therefore important from a religious and spiritual point of view (Figure 11). Local legends are told about its creation, as well as about the creation of the nearby large lake of Kyzylkol (meaning 'red angel') (importance for 10% of the participants) (Figure 11).

Finally, biodiversity and its maintenance ('maintaining nursery populations and habitats' [16], Figure 10) are distinguished by non-use values: biodiversity has an existence value ('characteristics or features of living systems that have an existence value' [16]), and represents a debt we owe to nature (33%), and a bequest value (14%) ('characteristics or features of living systems that have an option or bequest value' [16]) (Figure 11). It is biodiversity in general, fauna and flora everywhere: there is, therefore, no landscape frontier for these services.

## 4. Discussion

### 4.1. Identification of Ecosystem Services and Landscape Units

A large number of services and benefits were perceived by the stakeholders during interviews (Table 2). Among them, several CICES classes had not been identified in our initial literature review, e.g., 'characteristics or features of living systems that have an existence value'. Conversely, some ES found in the literature review were not perceived by the participants, e.g., 'hydrological cycle and water flow regulation'. It was thus necessary to rely on beneficiaries when identifying the priority ES, and to adapt the interviews according to the interviewees' answers. Therefore, the involvement of local stakeholders [7,11,24] and the consideration of all ES during the identification process is of utmost importance so that all priority ES are identified [22]. An example of important ES of drylands found in our literature review, but not perceived by the stakeholders was 'regulation of chemical composition of atmosphere and oceans' [25,26,47,49,51,53], that contributes to 'regulating our global climate' [16]. This ES can be hard to explain during interviews, and interviewees can have difficulties to link global ES and benefits to the local landscapes [30].

By using maps, the participants were able to refer to the majority of landscape units described in the literature for the provision of ES (Figure 3). The vision of local stakeholders is important for the identification of landscape units, even if in our case the units used were not different from those previously identified.

Stakeholder interviews confirmed that livestock farming is one of the main activities of the Sozak region, illustrating that people in drylands depend a lot on natural resources for their livelihood [25–28]. Indeed, reared animals, cultivated and wild plants, surface- and ground-water were major ES, identified by over 60% of interviewees, with almost all stakeholders citing pasture and grazed species when talking about wild plants related ES. Concerning regulation and maintenance services, they were less perceived by stakeholders, in agreement with Costanza et al. [14] (Table 2). Surprisingly, stakeholders were strongly and culturally attached to the biodiversity. The 'other biotic characteristics that have a non-use value' were an important group of ES, through 'characteristics or features of living systems that have an existence value' and 'characteristics or features of living systems that have an option or bequest value' services, (as they perceived that fauna and flora have as much right to exist as humans and should be preserved for future generations). In addition, in a study in Iran, Karimi et al. found a strong association between cultural services and biodiversity hotspots [23]. Such results contradict certain criticisms about the integration of ES into offsetting schemes. The concept of ES is controversial in biodiversity conservation [5], because it is considered difficult to link threatened resources or species with the concept of ES that have a strong utilitarian value and are provided in

landscapes dominated by humans [59]. However, we show that, in this very case study, the synergy between biodiversity preservation and cultural services could be exploited through biodiversity offset mechanisms, as also proposed by Sonter et al. [24]. Whether this relationship is generic or highly context-specific should be explored further.

*4.2. Exploiting the Link between Landscape, Ecosystem Services and Beneficiaries*

4.2.1. Suggestions of Several ES Selection Options: Impacted Ecosystem Services, Greatest Number of Beneficiaries or Common Landscapes

Biodiversity offset mechanisms should provide same services as those impacted during an economic development project (*like-for-like* or *in-kind* biodiversity offsets) to avoid creating or worsening social inequality [22]. Moreover, biodiversity offsets should focus on impacted beneficiaries, based on the number and characteristics of stakeholders, for example, targeting ES that impact the most beneficiaries, or those in a vulnerable category [22]. In addition, care must be taken to avoid bias from stakeholders wishing to maximize the benefits from specific services [5]. In view of the literature, we suggest several options for selecting ES to be integrated into biodiversity offset schemes.

**Option 1: like-for-like biodiversity offset.** We have shown that it is necessary to work with landscape units to identify which ES to incorporate into offset schemes (Figures 4–7). If like-for-like biodiversity offset was implemented in the Sozak district, services related to the landscape unit sand should be offset, as this is where direct impacts have taken place. Therefore, the following services would be prioritised (excluding mineral substance for energy, that was uranium): (i) 'wild plants grazed by reared animals', (ii) 'groundwater for drinking', (iii) 'control of erosion rate', (iv) 'characteristics of living systems that enable activities promoting health, recuperation or enjoyment through active or immersive interactions'.

As a development project can indirectly impact the provision of ES [5], indirect effects should also be considered in offsetting schemes (Figures 9–11). For example, by impacting winter pastures in the sand, mining can increase the use of cultivated plants for fodder, and so results in a greater use of mountain surface water for irrigation, leading to the negative effect of less surface water for drinking (Figure 9). Reducing winter pasture could also lead to a decrease in animal products. The negative effects of mining activities on wild plants also impacts the service maintaining nursery population and habitat and the associated cultural services (existence value and bequest value, Figure 11).

**Option 2: Biodiversity offset based on stakeholder categories and gender.** A selection based on beneficiaries could bring another point of view. Contrary to previous studies (e.g., [29,30]), we found no relationship between stakeholder category and gender with ES. This result is because almost all interviewees raised livestock (86%), even on a small scale for their own consumption, and those who did not consumed livestock products. Services such as 'animal reared for nutrition purposes' or 'wild plants grazed by reared animals' were therefore identified by almost all participants. Similarly, services related to biodiversity and its maintenance were identified by more than three quarters of interviewees. Therefore, in this type of rangeland, targeting ES through biodiversity offset will not lead to a bias in stakeholder category or gender inequality, nor will the selection of ES based on vulnerable categories of stakeholders be necessary (such as women or elders [22]).

**Option 3: Biodiversity offset based on common landscape units.** Our results suggest that from the point of view of equality between beneficiaries across the landscape, offsetting could target ES produced by the four most common landscapes identified by interviewees (Figure 8), i.e., sand, steppe, village and no landscape frontier. If we add a criterion of targeting ES that impact the most beneficiaries [22], we are able to prioritise ES among these four landscape units. We suggest that by cross-referencing the ES related to the four relevant landscape units (Figures 5–7) and targeting ES identified by most interviewees (Figures 9–11), we should prioritise those ES in the following types of landscape:

(i)     Sand: 'Wild plants grazed by reared animals' (Figures 5 and 9) and 'Control of erosion rate' (Figures 6 and 10)

(ii)  Steppe: 'Cultivated plants for nutrition', 'Cultivated plant for fodder', 'Material from wild plants' (medicinal use) (Figures 5 and 9) and 'Control of erosion rate' (Figures 6 and 10)

(iii)  Village: 'Reared animals for nutrition', 'Reared animals for energy' (manure), 'Groundwater for irrigation' (Figures 5 and 9), 'Decomposition and fixing processes and their effect on soil quality' (Figures 6 and 10), and 'Characteristics of living systems that enable aesthetic experiences' (Figures 7 and 11).

(iv)  No landscape frontier: 'Maintaining nursery population and habitat', 'Seed dispersal' (wind-induced) (Figures 6 and 10), 'Characteristics or features of living systems that have an existence value' and 'Characteristics or features of living systems that have an option or bequest value' (Figures 7 and 11).

### 4.2.2. Choice of Sites for Biodiversity Offset Schemes

Since ES are associated with beneficiaries, off-site biodiversity offset is not an option. In addition, to provide benefits to all stakeholders, common landscape units should be used for offset (Figure 8). For example, if offsetting was implemented around the Shu river or Betpak-Dala steppe, northern villages would be favoured by ES, whereas southern villages would benefit more from offsetting in the mountains, foothills and lakes area. Therefore, the units sand, steppe, village and no landscape frontier should be used for biodiversity offset schemes.

### 4.3. From Suggestions to Biodiversity Offset Strategies: What to Offset and Where?

By cross-referencing options 1–3 above (Section 4.2.1) with site selection (Section 4.2.2), we found different strategies. (i) The first would prioritises the ES 'wild plants grazed by reared animals' and 'control of erosion rate' in the sand landscape unit. This strategy would be a like-for-like offset solution, as the ES targeted are those that are potentially impacted in the *sand* unit (Figure 5) where the uranium mines are located. Furthermore, biodiversity offsets would be implemented in the landscape unit that initially provided the ES (Figure 5) and is a unit common to all villages (Figure 8). If such an offsetting strategy was not possible to implement, other possibilities would be: (ii) like-for-like offsets but in another landscape unit, such as steppe, that provides similar ES (Figure 5). (iii) Use of the no landscape frontier unit, where suitable solutions to prioritise could be protection measures for the preservation of fauna and flora (Figure 6), contributing to the existence value and bequest value of biodiversity-related ES (Figure 10), that are indirectly impacted by mining activities. As ES provided by no landscape frontier units can be perceived anywhere, protection measures could be implemented in common landscape units such as sand or steppe (Figure 8). However, the site should be chosen in terms of its ability to compensate for biodiversity and it would have to be verified that sufficient ecological gains can be achieved in the steppe or sand landscape units [22]. (iv) *Out-of-kind* offsets (meaning that the ES targeted are not those impacted), such as the planting of trees in villages, would contribute to the service 'characteristics of living systems that enable aesthetic experiences' and reach most of the beneficiaries (Figure 11).

### 4.4. Service-Based Scenarios and Their Potential Indirect Impacts on Other Provision of ES

Since offsetting schemes themselves can have indirect impacts on the provision of ES [10,11], we can use Figures 9–11 to consider those impacts. (i) and (ii) Ecosystem services such as 'control of erosion rate' could be targeted through the protection and restoration of vegetation [60]. However, such scenarios would initially reduce winter grazing and associated services (wild plant grazed by reared animals and then reared animals for nutrition, material and energy) until the vegetation is re-established (Figure 9). (iii) The implementation of biodiversity protection measures would improve the service 'maintaining nursery population and habitat', and the associated cultural services 'characteristics or features of living systems that have an existence' value and 'characteristics or features of living systems that have an option or bequest value' (Figure 11). Nevertheless, such measures would lead

to a decrease of other ES [10,11], either through access to winter pastures if implemented in the sandy desert, or access to medicinal plants in the steppe (Figure 9). (iv) Planting in villages to green and beautify would improve the service 'characteristics of living systems that enable aesthetic experiences' (Figure 11) but would also involve higher consumption of groundwater for irrigation, which would then be less available for vegetable gardens and orchards (Figure 9). Another example of an indirect impact through biodiversity offset, exists already in the Sozak district: saxaul trees are planted in compensation schemes, but are protected and cannot be used as firewood (Figure 9) without the permission of the forest authorities. Therefore, the service 'wild plants for energy' is reduced.

### 4.5. The Benefits of Considering Ecosystem Services in Biodiversity Offsets

Populations in rangelands rely heavily on ES for their livelihoods [5,24]; therefore, biodiversity offset mechanisms should also compensate for the ES impacted. Our study does not call this into question. We provide a framework for identifying the ES to prioritise for beneficiaries in the right landscape units according to the vision of the local stakeholders. This approach highlights the importance of implementing offset schemes in the landscapes closed to the area impacted, thus providing a fairer offset for the beneficiaries [5]. As offset mechanisms can affect ES or access to ES [10,11,24], we suggest that a systemic approach to ES identification (i.e., at the landscape scale) should be implemented during the offset scenarios' design stage. This approach would avoid the potential indirect impacts sometimes caused by biodiversity offsets.

### 4.6. Towards a Framework Applicable Worldwide

The developed framework incorporates recommendations from the literature on the integration of ES in biodiversity offset schemes. We propose a systemic approach [22], which considers ES at the landscape scale in order to identify priority ES, access to these different ES by beneficiaries [11] and also to improve understanding of impacts on social needs and preferences [22]. In their study, Souza et al. state that a review of ES is necessary for ES-oriented offsets [11]. Our framework addresses the first two steps of this review: the identification and prioritisation of ES.

The developed framework could be used worldwide for future offset plans. Figure 2 shows the methodology to be applied. The framework is intended to be generic but there are specific parts for each area worldwide and new context: (i) the literature search on ecosystem services to adapt the interview guide to other study areas; (ii) the determination of stakeholder categories, as they should represent the diversity of local stakeholders; (iii) the choice of villages (or settlements, e.g., in the case of mobile populations); and (iv) the Google Earth and Google Map background maps used during the interviews. Through this framework, we test some recommendations for the prioritisation step: targeting ES lost in development and targeting the most beneficiaries [22]. Nevertheless, in future research and other area of the world, other prioritisation criterions can be used. As shown in Figure 2, depending on the local context, other questions and hypothesis can be tested, in addition to the relationship between landscape and ES and stakeholders and ES ('Other possible combinations depending on the questions to be tested for the ecosystem services provision').

### 4.7. Ecosystem Services-Based Offsets as a Complementary Measure in Biodiversity Offset

In order to effectively integrate ES into biodiversity offset schemes, other elements have to be assessed. Planning offsets for both biodiversity and ES can be challenging. Areas providing ES and ecologically important areas are not always compatible [5,11]. Offset strategies targeting impacted ES may, therefore, not meet the criteria for biodiversity offsets [11]. Furthermore, they may not fulfil the legal requirement. For example, in Kazakhstan in the case of ISR mining 'Subsoil users, when using plots of the state forest fund for uranium mining by the method of underground borehole leaching, shall be obliged, during the first three years of subsoil development, to make compensatory plantings of

forest plantations in double the size of the area' (Article 54 of the Forest Code of the Republic of Kazakhstan [61]). However, decisions on the modalities of the plantation are taken at the level of regional authorities [61]. Thus, in order to achieve ecological and biodiversity objectives and meet legal requirements, biodiversity offsets for ES should take the form of additional or complementary measures [5,11].

The implementation of additional measures for ES would avoid some drifts. Indeed, it would be risky to supplant the current approach of biodiversity offsets, based on species and habitats: an approach based only on ES may lead to the substitution of the original species or habitats by other species and habitats providing the same ES as those impacted [5]. Therefore, Jacob et al. [5] propose that the integration of ES should be conducted in a second step, after ensuring that the ecological equivalence required by biodiversity offsets is achieved.

When offsetting ES, the potential biodiversity offset schemes should be differentiated according to the categories of ES considered. Provisioning and cultural ES should be accessible to communities, whereas this is not necessarily the case for regulating and maintenance services [11]. Attention should be paid to increasing access to provisioning services, that can sometimes negatively impact cultural and regulation and maintenance services [5]. When a development project is implemented, it should also be taken into account that some ES are simply not compensable, for example, a unique area that provides important spiritual or aesthetic ES for populations will most likely never be compensated for if lost [5].

## 5. Conclusions

We developed a systemic approach to integrate ecosystem services (ES) into biodiversity offset schemes to compensate for the negative impacts of economic development projects. We outlined a framework that allowed ES to be identified and prioritised across different landscapes. Interviews with local stakeholders allowed us to determine bundles of ES that impacted as many beneficiaries as possible. Interviewees also efficiently described the landscape units providing those ES. The category (local authority, elder, herder farmer, social and health worker, mother with many children, teacher or inhabitant) and gender of participants did not influence the identification of ES in this specific context of Kazakh rangelands. Stakeholders preferred services provided by the landscape units close to their village. Since ES and landscape units are significantly linked, we suggest that biodiversity offset should target ES provided by the landscape where mining activities occur. We show that to avoid conflict and bias when offset schemes are implemented, ES provided by landscape units accessible to all villagers should be targeted as a priority. In addition, a systemic understanding of the provision of ES across different landscape units would make it possible to consider both the potential direct and indirect impacts of development project and biodiversity offset scenarios on ES.

**Supplementary Materials:** The following supporting information can be downloaded at: https://www.mdpi.com/article/10.3390/land12010202/s1, Table S1. Interview guide for identification of ecosystem services. Table S2: *p*-value of the 13 $\chi^2$ test performed on the ecosystem services and landscape units matrix tables. A *p*-value $< 0.05$ rejects the H0 hypothesis (independence between ecosystem services at each level of CICES and landscape unit) and shows that the two variables are related. Table S3: Links between ecosystem services class (C) (or group (G) when class contribution to the axes is not sufficient) and landscape units, according to the respective contributions of each variable to dimensions (axes) and their relative position on the contributed axes of the performed correspondence analysis. Table S3 summarizes the results of 6 correspondence analysis (Group of provisioning ecosystem services and landscape units; group of regulation and maintenance ecosystem services and landscape units; group of cultural ecosystem services and landscape units; class of provisioning ecosystem services and landscape units; class of regulation and maintenance ecosystem services and landscape units; class of cultural ecosystem services and landscape units).

**Author Contributions:** Conceptualization, A.B., A.I. and J.-D.C.; methodology, A.B., A.I. and J.-D.C.; formal analysis, A.B.; investigation, A.B., A.I. and S.K.; data curation, A.B.; writing—original draft preparation, A.B.; writing—review and editing, A.B., A.I., J.-D.C., P.S., V.R. and S.T.; visualization, A.B.; supervision S.T., V.R. and P.S.; project administration, S.T. and V.R.; funding acquisition, V.R. and S.T. All authors have read and agreed to the published version of the manuscript.

**Funding:** This research was funded by Orano Mining, France.

**Data Availability Statement:** The data presented in this study are available on request from the corresponding author.

**Acknowledgments:** We would like to thank all the participants who agreed to take part in the interview, the employees of the akimats in the eight villages for organising the interviews and welcoming us to their offices, KATCO's CSR department for organising the interviews prior to the fieldwork, our Kazakh-French interpreter Akbope Saparaliyeva, KATCO for the logistics, support and assistance provided in Kazakhstan, Alexia Stokes (native English speaker) who proofread the article, and Orano for granting permission to publish our results. We also thank the anonymous reviewers and editors for their valuable comments and suggestions.

**Conflicts of Interest:** Véronique Rayot (Orano Mining) reviewed the article as a co-author (contribution in the Writing, review and editing section). Sholpan Koldasbekova (KATCO JV LLP) worked with Annaêl Barnes in the selection of villages and categories of stakeholders (i.e., local authority, elder, herder-farmer, social and health worker, mother of a large family, teachers and villagers). She was in charge of contacting the mayors or mayors' assistants of each village to schedule the interviews. (Contribution in the investigation section).

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
