# Peer review of "Improving Biodiversity Offset Schemes through the Identification of Ecosystem Services at a Landscape Level"

_land, doi:10.3390/land12010202_

Round 1
Reviewer 1 Report
The authors developed a framework for the identification of ES for their integration in biodiversity offsets. This framework
was tested in rangelands landscapes in Kazakhstan, in a context of uranium mining.
1. Please highlight the key points in the abstract.
2. How come to Stakeholder's categories?
3. It is recommended to add specific method description.
4. The results should be better described, discussed and justified.
Reviewer 2 Report
The authors developed a framework to identify integration of ecosystem services (ES) in biodiversity offsets, and tested the framework in rangelands landscapes in Kazakhstan, in a context of uranium mining. The authors found a significant link between ES and landscape units, but there were not any significant links between ES and stakeholder categories or gender. The study is interesting, however, there are problems with the manuscript which would need a substantial revision.
Some suggestions are as follows:
1. Introduction. It’s better to clarify the goal of this study. Relationships between ES and biodiversity as well as between ES/biodiversity offsets and stakeholder should be strengthened in this section.
2. Materials and Methods. It is better to add more information on the data used in the study, such as references and/or link. The authors should improve their descriptions of hypothesis.
3. Results. Line 334: where is the hypothesis? Figure/Table descriptions (e. g. lines 340-343 and section 3.2) should not be written as a single paragraph.
4. Discussion. More information about mechanism between ES and biodiversity offsets should be discussed in the section. Discussions about the applicable of the framework in other areas worldwide should be strengthened.
5. The language and writing need to be improved. A lot of sentences are too long to be clearly read. The structures of results and discussion is confusing to read.
Reviewer 3 Report
Rev LAND
This a very interesting and complete manuscript about a research carried out in a highly critical context (uranium mining) belongings to a heterogeneous country of high ecological value, poor studied in this regard. I read it with great interest. Style and logic are clear. I like this type of manuscript where knowledge and research are strictly connected to a pragmatic approach: ES and EIA are both topics strategic in environmental planning and conservation. Language seems good (but I am not Mother Tongue in English). I think that this ms deserves to be published as soon as possible after very MINOR REVISIONS: this types of papers are wellcome. I suggest below only few points.
Row 56. There is white double space between words.
Fig. 1. Good and well-readable.
row 138, 141, 443 and everywhere. ‘spp.’ Not in italic font (is for’ species’).
Row 153: ‘23 people km²’ should be ‘23 people/km²’
Row 286. Change ‘chi²’ with the Greek symbol (χ²). Also in row 386 and everywhere.
Figures 5,6,7 are very original and useful. Fig. 9, 10 and 11 are very useful figure deserving publication on a handbook on this topic (ES)!
Rows 411, 577. Not correct the ‘Figure 9Figure 10Figure 11).’. See Instructions for authors.
I like to re-review a new version.
Add the role of anonymous reviewers and Editors in Acknowledgments.
Have a nice work.
Round 2
Reviewer 2 Report
I was satified with the revised manuscript.